# A Retrospective Descriptive Study of *Staphylococcus* Species Isolated from Canine Specimens Submitted to a Diagnostic Laboratory in South Africa, 2012–2017

**DOI:** 10.3390/ani14091304

**Published:** 2024-04-26

**Authors:** Themba Sigudu, Daniel Qekwana, James Oguttu

**Affiliations:** 1Department of Agriculture and Animal Health, College of Agriculture and Environmental Sciences, University of South Africa, Johannesburg 1709, South Africa; joguttu@unisa.ac.za; 2Department of Health and Society, School of Public Health, Faculty of Health Sciences, University of Witwatersrand, Johannesburg 2193, South Africa; 3Section Veterinary Public Health, Department of Para Clinical Sciences, Faculty of Veterinary Science, University of Pretoria, Pretoria 0110, South Africa; nenene.qekwana@up.ac.za

**Keywords:** *Staphylococcus*, coagulase negative, CoNS, coagulase positive, CoPS, staphylococci, canine, isolates

## Abstract

**Simple Summary:**

The present study addresses the lack of research on *Staphylococcus* spp. among dogs in South Africa. The study characterises isolates from retrospective data from a veterinary diagnostic laboratory in terms of time, place, and animal factors. Using data collected from 2012 to 2017, the study analysed 1627 positive *Staphylococcus* isolates and identified 10 species, the majority (92.0%) of which were classified as coagulase-positive, while a few were coagulase-negative (6.0%) and coagulase-variable (3.0%). Male dogs contributed just over half (53.2%) of the isolates. Dogs aged ≥9 years contributed the most isolates (23.2%). KwaZulu-Natal Province contributed the majority (45.0%) of the isolates, while Northern Cape Province contributed the least (0.1%). Almost half (46.0%) of the isolates came from skin specimens. The study demonstrated a limited variation in the number of *Staphylococcus* isolates across seasons and the occurrence of a diversity of *Staphylococcus* species among dogs in South Africa. There is a need for research to improve our understanding of factors that influence the observed disparities in *Staphylococcus* spp. Proportions observed in this study.

**Abstract:**

There is a scarcity of published studies on the occurrence of *Staphylococcus* spp. Among dogs in South Africa. The objective of the study was to characterise the *Staphylococcus* spp. Isolated from dog samples submitted to a veterinary diagnostic laboratory in South Africa in terms of time, place, and person. This study utilised a dataset of 1627 positive *Staphylococcus* isolates obtained from a veterinary diagnostic laboratory in South Africa from 2012 to 2017. Out of the 1627 confirmed isolates, 10 different species of *Staphylococcus* were identified. Among these, 92.0% were classified as coagulase-positive staphylococci (CoPS), 6.0% were coagulase-negative staphylococci (CoNS), and 3.0% were coagulase-variable. Male dogs contributed just over half (53.2%) of the *Staphylococcus* isolates, while female dogs contributed the remaining 46.8%. The largest proportion of isolates (23.2%) were obtained from dogs aged ≥ 9 years, with the highest number of isolates originating from KwaZulu-Natal Province (45.0%) and the least from Northern Cape Province (0.1%). Out of the total samples included in the records, the majority (46.0%) were skin specimens. The number of *Staphylococcus* isolates recorded showed limited variation between the seasons (24.3% in autumn, 26.3% in winter, 26.0% in spring, and 24.0% in summer). This study highlighted the diversity of *Staphylococcus* spp. isolated from dogs, and the burden of staphylococcal carriage among dogs in South Africa. Further research is required to examine the factors that contribute to the observed discrepancies in the proportions of *Staphylococcus* spp. between the provinces.

## 1. Introduction

*Staphylococcus* is a facultative anaerobic, Gram-positive, coccus-shaped bacteria that occurs in clusters [1]. These organisms are frequently isolated from the skin and mucous membranes of dogs as commensal organisms. Even so, they may cause diseases such as otitis externa and pyoderma in canines [2]. Infection typically develops when the skin or mucosal barrier is compromised due to underlying factors such as medicinal and surgical interventions, atopic dermatitis, or immunosuppressive conditions [3]. There are forty-three known *Staphylococcus* spp. that can be divided into two broad categories, namely, coagulase-positive staphylococci (CoPS) and coagulase-negative staphylococci (CoNS) species, based on their coagulase enzyme production, which causes blood clotting [4]. Even though several CoNS, including *S. epidermidis* and *S. haemolyticus*, have been isolated from dogs, it is believed that these commensals are less or non-pathogenic [5]. On the other hand, CoPS include *S. pseudintermedius*, which is the main staphylococcal pathogen that affects dogs [6]. The prevalence of *Staphylococcus* infection in dogs has been recognised as an increasing concern in the field of veterinary medicine [7]. They have a substantial impact on the occurrence of skin and surgical site infections [8] and present considerable treatment difficulties [9]. Furthermore, dogs can serve as a likely source of *Staphylococcus* aureus-related infections or re-infections in humans [8]. There is compelling evidence of the transmission of these organisms between humans and animals [10]. This makes it an important health and/or public health problem due to the close companionship between dogs and humans. The seriousness of the problem is appreciated when consideration is given to there being 9 million pet dogs within the boundaries of South Africa [11]. *Staphylococcus* species have been associated with several clinical diseases in dogs, such as pneumonia, nephritis, sepsis, otitis, pyoderma, post-surgical infections, wound infections, and nasal infections [12,13]. Most infections, though, have been associated with canine dermatological problems. The majority of *Staphylococcus*-associated infections in canines are caused by *Staphylococcus aureus* and *Staphylococcus pseudintermedius*, owing to the high rates of colonisation observed in dogs with these particular species. All types and sizes of veterinary facilities experience hospital-associated infections (HAIs), and this frequency is anticipated to rise. In the field of veterinary medicine, pneumonia, urinary tract infections, bloodstream infections, infectious diarrhoea, and surgical site infections are commonly observed HAIs [14]. Nevertheless, research has indicated that dog bite wounds provide a potential risk for infection. A study conducted in the UK found *S. pseudintermedius* in 6 out of 34 bite wounds [15]. Humans can experience transient colonisation while treating dogs with profound pyoderma caused by MRSP. According to findings in a study by Abdullahi et al. [16], the most probable origin was contact with purulent material. Therefore, exposure to this and contact during bathing may constitute risk factors for human colonisation. At present, there are a limited number of published research papers with regard to the prevalence of *Staphylococcus* spp. among dogs in South Africa. The primary focus of existing investigations has been on infections caused by *Staphylococcus aureus* [15,17]. There is a need to enhance our comprehension of the prevalence and impact of infections caused by a variety of *Staphylococcus* species in developing countries like South Africa. This has the potential to enhance the identification of *Staphylococcus* infections and offer valuable insights for empirical treatment. The aim of the investigation was to analyse *Staphylococcus* spp. strains obtained from dog samples submitted to a veterinary diagnostic laboratory in South Africa from 2012 to 2017 in terms of time, place, and animal attributes.

## 2. Materials and Methods

### 2.1. Study Design and Setting

The current study employed a cross-sectional retrospective research design to accomplish its goals. This study included laboratory data from 1627 samples from dogs that were submitted to a veterinary diagnostic laboratory between January 2012 and December 2017. The laboratory caters to the private veterinary healthcare sector in South Africa and offers information to veterinarians and the livestock industry on the prevention and treatment of diseases that are of veterinary significance for various animal species, including companion animals, livestock, wildlife, poultry, and equines. The specimens submitted for microbiological investigation included samples from various bodily regions, such as the ear, skin, hair, nails, urine, mucosal surfaces, and other body sites.

### 2.2. Data Extraction

Data on all *Staphylococcus* species recorded from 2012 to 2017 were extracted from the database of a single veterinary diagnostic laboratory. Demographic (age and sex), spatial (province of origin), clinical (species of organism and specimen type), and temporal (season) information were extracted for each isolate.

### 2.3. Isolating Staphylococcus Species

To isolate *Staphylococcus bacteria*, samples were cultured on blood agar and eosin methylene blue agar plates, and then incubated at 37 °C with 5–10% CO_2_ for at least 24 h. In cases where contamination was anticipated, such as with nasal swabs, cultures were also grown on Columbia colistin and nalidixic acid (CNA) plates supplemented with blood. After the initial incubation, the plates were checked for the presence of pathogenic bacteria. They were then further incubated for another 24 h at 37 °C under aerobic conditions and rechecked for pathogenic growth. Microorganism presence was reported based on either isolation in pure culture or being the predominant microorganism in the specimens.

### 2.4. Identifying Staphylococcus Species

*Staphylococcus* isolates were identified by typical colony morphology, positive catalase, and negative oxidase reactions. Species-level identification was conducted using DNAse agar instead of coagulase tests and assessing maltose acid production. *Staphylococcus aureus* was identified as DNA- and maltose-positive, while *Staphylococcus pseudintermedius* was DNA-positive and maltose-negative.

### 2.5. Coagulase Activity Testing

Coagulase activity was tested using the tube coagulase test, where rabbit plasma was inoculated with staphylococcal bacteria colonies that were catalase-positive Gram-positive cocci. The tube was then incubated at 37 °C for around 1.5 h, with further incubation up to 18 h for negative results. Positive results, indicative of organisms like *S. aureus*, were characterised by plasma clot formation. Negative results, suggesting organisms like *S. epidermidis*, indicated liquid plasma. However, confirmation tests such as biochemical assays were necessary for definitive identification, as false negatives could occur due to inadequate cooling.

### 2.6. Data Management

The data underwent a thorough examination to identify any inconsistencies, such as missing information, incorrect addresses, and duplicate records, before analysis. No instances of duplication were detected, and there were no reported cases of mixed infections. The variable “age” was re-categorised into five distinct categories: <2 years, 2–4 years, 5–6 years, 7–8 years, and >8 years, as outlined in the literature [18]. The specimen type was categorised into six distinct groups: ear, skin, hair and nails, urine, mucosal surfaces, and “Other sites”. The specimens labelled as ‘Other sites’ consisted of unidentified specimen types and types that accounted for less than 0.02% of the overall number of isolates. The months of the year were classified into four distinct seasons: autumn (March, April, and May); winter (June, July, and August); spring (September, October, and November); and summer (December, January, and February).

### 2.7. Data Analysis

Data processing and analysis were conducted using Stata Statistical Software version 17 [19]. Crude and factor-specific proportions for categorical variables, along with their corresponding 95% confidence intervals (95% CI), were computed. These calculations were based on the variables time, person, and place. Temporal graphs were used to depict the annual variation in the proportion of *Staphylococcus* spp.

## 3. Results

Between 2012 and 2017, a total of 1627 *Staphylococcus* isolates were identified. Out of these, 10 different *Staphylococcus* species were recorded, as shown in Table 1. Of the 1627 isolates recorded, the overwhelming majority (91.8%) were CoPS, while 5.8% were CoNS and 2.5% were classified as coagulase-variable (i.e., exhibited both coagulase positive and coagulase negative characteristics). Overall, *S. pseudintermedius* was the most commonly isolated, accounting for 86.0% of the species in the data set, followed by *S. aureus* (5.84%), while *S. epidermidis* made up only 5.16% of the total number of isolates.

### 3.1. Distribution of Staphylococcus spp. Based on the Age and Sex of the Dogs

Based on age, there was a minimal variation in the distribution of the different *Staphylococcus* spp. observed among the different age groups, with the age group ≥9 years having the lowest number of CoPS (89.1%) in comparison to other age groups (Figure 1). On the other hand, compared to other age categories, the percentage of CoNS was most elevated among canines that were ≥9 years old (8.2%), while the age group <2 years had the lowest number of CoNS staphylococcal isolates (3.8%) in relation to other categories based on coagulase activity.

Overall, a slight majority of the isolates (53.2%) were obtained from male dogs, with the remaining 46.8% obtained from female dogs. Although the male dogs contributed a slightly higher percentage (92.5%) of CoPS isolates, there was a minimal difference between the number of CoPS isolated from both female and male dogs (Table 2). Likewise, there was a slight variation in the number of CoNS isolates obtained from females compared to males. However, with CoNS, female dogs contributed a slightly higher number (6.6%) compared to males (5.1%). There was also a slight variation between the coagulase-variable *Staphylococcus* species obtained from female (2.5%) and male (2.4%) dogs.

The CoPS remained the predominant species, constituted the majority of isolates recorded each year, and showed minimal variation throughout the study period. Meanwhile, the number of CoNS and coagulase-variable species recorded each year showed variation over the study period. For example, the number of CoNS isolates varied between 1.4 and 9.2%, while the Coagulase-variable *Staphylococcus* varied from 0% in 2012 to 5% in 2013.

Based on season, there was a slight variation in the distribution of isolates across the seasons, with autumn accounting for 24.2%, winter for 26.2%, spring for 25.6%, and summer for 24.0%. While CoPS were the most isolated, there was minimal variation in the numbers recorded across the four seasons. Likewise, the number of coagulase-variable *Staphylococcus* isolates was consistent over the four seasons. With regard to CoNS, the lowest number of isolates was recorded in the summer (2.6%), but otherwise there was minimal variation across the rest of the seasons (Table 2).

The skin specimen contributed most *Staphylococcus* spp. (50.3%) isolates, and most of these isolates were CoPS (80.4%), followed by CoNS (14.7%). The ear specimen contributed the second highest proportion of isolates (25.0%), and likewise, the majority of isolates from ear specimens were CoPS (89.1%). The urinary specimen contributed 11.0% of the total isolates, and most of these isolates were CoPS (73.4%). Respiratory specimens accounted for only 5.0% of the total isolates, but the majority of the isolates from the respiratory system were CoPS (70.7%) (Table 2).

With regard to the temporal trends, as shown in Figure 2 below, the number of *S. pseudintermedius* isolates observed rose from 125 in 2012 to 372 in 2017, demonstrating a consistent increase in the number isolated over the duration of the study. On the other hand, the number of *S. aureus* isolates recorded fluctuated from year to year, with the highest number observed in 2017 (29 isolates) and the lowest in 2013 (8 isolates). But from 2012 to 2017, there was a slight overall increasing trend in the number of isolates observed.

The number of *S. epidermidis* isolates showed a decreasing trend from 2012 to 2015 and then a drastic increasing trend until 2017. The number of unspecified *Staphylococcus* species (other species) varied considerably from year to year, showing an initial increase between 2012 and 2013, a slight decrease from 2014 to 2015, followed by a slight increase between 2015 and 2016, before decreasing again between 2016 and 2018 (Figure 2).

Overall, *S. pseudintermedius* was consistently the most isolated *Staphylococcus* species throughout the study period. This was followed by *S. aureus* and *S. epidermidis.*

In terms of the spatial distribution, KwaZulu-Natal recorded the highest number of *Staphylococcus* species, comprising nearly 45% of the total isolates. This was followed by the Western Cape (25%) and Gauteng Provinces (17%). Mpumalanga, Eastern Cape, and Limpopo Provinces contributed moderately low numbers of isolates (7.6%, 3.8%, and 1.6%, respectively). The Free State, Northwest, and Northern Cape had the lowest numbers of isolates, each Province contributing less than 1% to the total (Figure 3).

### 3.2. Correlation between Age, Sex, Season, and Specimen Type

Based on the observed *p*-values for sex (*p* = 0.577) and season (*p* = 0.220), there were no statistically significant differences in the distribution of specimen types across different sexes and seasons. However, the *p*-value for the chi-square test for age (*p* = 0.087) showed that there was a marginally significant association between age and specimen type (Table 3).

## 4. Discussion

This study detailed the distribution and burden of *Staphylococcus* infection in dogs whose samples were submitted to the veterinary diagnostic laboratory. The study also highlights the temporal and spatial distribution of staphylococcal infections in dogs in the study area.

The prevalence of *Staphylococcus* species is a growing and pressing concern in both veterinary and public healthcare settings [20]. This investigation offers valuable insights into staphylococcal carriage among dogs in South Africa, which might serve as the basis for future research. The established epidemiological association between *S. aureus* infection in humans and animals is widely recognised [19,21,22,23,24] and the infection of humans by *S. pseudintermedius* has also been documented [25]. Therefore, the occurrence of these two organisms in dogs, as demonstrated in this study, suggests that dogs are a potential source of this organism for humans.

Available evidence suggests that although *S. aureus* is more commonly found in humans than animals, the opposite is true for *S. pseudintermedius*. The later is more prevalent in dogs than in humans. This is confirmed by research conducted by Donoghue and Boost [26] in 2004 and Teixeira et al. [27,28] in 2009 in Japan, Canada, and Hong Kong. The occurrence of *S. pseudointermedius* in dogs varies between 61% and 89.5%, while the occurrence of *S. aureus* in dogs can vary from 9% to 40%. Likewise, the results of our study revealed that *S. pseudintermedius* (86.0%) is the most commonly isolated in dogs relative to other species. This is confirmed by Teixeira et al. [27], who also noted that *S. pseudintermedius* was the most prevalent in dogs in comparison to other species. This could be elucidated by the fact that *S. aureus* is not typically a commensal organism in dogs.

The findings of the present study showed that dogs in South Africa carry various *Staphylococcus* spp., such as *S. saprophyticus*, *S. chromogenes*, and *S. lentus*. This is consistent with what has been reported by other authors in studies done elsewhere. For example, it has been proven that canines can contract different *Staphylococcus* species, including *S. schleiferi*, *S. epidermidis*, *S. xylosus*, and *S. felis* dogs [27].

*Staphylococcus pseudintermedius* emerged as the most prevalent *Staphylococcus* species in dogs, and there was a consistent increase in the number of isolates of this species over the study period. In addition, there were also fluctuations in the number of *S. aureus* and *S. epidermidis* across the years. The observed trends highlighted the dynamic nature of *Staphylococcus* species and underscored the importance of continued surveillance and monitoring efforts. This could help shed light on the causes of the observed fluctuations.

The CoPS group constituted the majority (91.8%) of species isolated throughout the study period. These findings are consistent with the results of several other studies conducted elsewhere, such as those by Perreten et al. [29] in 2010, Lilenbaum et al. [30] in 2007, and Vanni et al. [31] in 1999, that reported high percentages of a CoPS species, *Staphylococcus pseudintermedius*.

CoNS were observed in this study, albeit less frequently. Among the CoNS species isolated from canine specimens, *Staphylococcus epidermidis* (5.2%) emerged as the predominant one. This finding suggests that S. epidemidis, typically considered a commensal bacterium of the skin and mucous membranes, may act as an opportunistic pathogen in dogs, particularly in the context of medical device-associated infections.

The higher number of *Staphylococcus* species isolated from male dogs compared to females in this study suggests a heightened propensity for male dogs to acquire staphylococcal infections as compared to their female counterparts. A similar trend has been observed in several human studies that have reported that males, as compared to females, tend to be at a greater risk of being infected by *Staphylococcus* [4,25,32]. Scientists attribute hormonal and behavioural variations between sexes to the differences in the staphylococcal colonisation observed in humans. However, no explanation could be found for the difference observed between male and female dogs in this study.

Although the present study did not demonstrate clear variations in the number of *Staphylococcus* isolates over the four seasons, other studies have shown a higher occurrence of *Staphylococcus* infections during the summer and autumn months, or warmer periods. This has been noted consistently across various studies. For example, investigations into the seasonal trends of *S. aureus* skin and soft tissue infections (SSTI) conducted in a tropical region underscore the influence of local humidity and inadequate personal hygiene as significant factors contributing to *S. aureus* colonisation and infection [18]. Another study revealed notably larger bacterial populations on the back, axillae, and feet of individuals in environments with high temperatures and humidity compared to those in environments with moderate temperatures and low humidity [19]. The difference observed between the results of human studies and this one could be due to the fact that season does not play a role in the colonisation of dogs by *Staphylococcus* species.

KwaZulu-Natal province had the highest number of isolates compared to other provinces. The reasons behind this observation remain unclear. However, the authors of this study suggest that the elevated count of *Staphylococcus* isolates from KwaZulu-Natal Province, in contrast to other provinces, might be linked to the diagnostic laboratory’s significant operation primarily in KZN from 2012 until April 2016. This period saw the opening of the Gauteng laboratory, and microbiology previously conducted by the Cape Town branch was integrated with Gauteng. We also hypothesise that provinces have different geographical features, climates, and ecosystems that may harbour varying populations of *Staphylococcus* species. Available evidence suggests that environmental conditions such as temperature and humidity can affect bacterial growth and survival, and thus *Staphylococcus* species may adapt to local environments and host populations, resulting in differences in prevalence and distribution patterns observed across provinces [33,34,35,36,37]. However, a study conducted in Europe found that *Staphylococcus* infection exhibited characteristic epidemic patterns and was mostly concentrated in specific regions [30]. The varying degree of distribution of *Staphylococcus* isolates across the nine provinces could reflect differences in population density, healthcare infrastructure, or environmental factors influencing bacterial prevalence. There are some studies conducted in the southern hemisphere [38,39]. Generally, their findings paralleled those of studies conducted in the northern hemisphere concerning similar types of infections. They showed summer peaks for skin and soft tissue infections and no discernible seasonal variation in peritoneal dialysis-related peritonitis [39,40]. Other studies from tropical and subtropical regions noted seasonal variation in *S. aureus* infections, and among these, only one indicated peaks during summer or warm weather for SSTI [39,41]. In one study from the tropical Northern Territory of Australia, where temperatures remain relatively stable throughout the year, the highest incidence of *S. aureus* infection occurred during the wet season [38].

Understanding regional differences in *Staphylococcus* prevalence can inform public health strategies, including surveillance, infection control measures, and antimicrobial stewardship efforts tailored to specific geographic areas. As a result, further research may be warranted to explore the factors contributing to the observed variations in *Staphylococcus* prevalence among different provinces in South Africa.

Overall, the results of the present study did not demonstrate evidence of an association between specimen type and the variables of sex, age, or season. However, a weak association was observed between the age and type of specimen from which *Staphylococcus* were isolated. However, a study conducted in the United States of America revealed statistically significant associations between specimen type and demographic variables in humans, suggesting that sex, age, and season significantly influence the choice of specimen type in infectious disease cases [42].

## 5. Limitations of the Study

The data included in this investigation did not differentiate between contaminants and infections caused by CoNS species. In addition, the data utilised in this study did not provide information on whether the canines from which the samples were taken were inpatients or outpatients. The lack of these variables limited the analysis that could be performed using the data. Furthermore, in April 2016, the KZN microbiology laboratory was closed, and the Gauteng laboratory started operations with an entirely new personnel complement. In view of this, caution should be exercised when interpreting the results reported in this study. This notwithstanding, the findings of this study offer valuable preliminary information on the occurrence of *Staphylococcus* species among canines presented at private veterinary healthcare facilities in South Africa.

## 6. Conclusions

Various species of *Staphylococcus* are associated with dogs in South Africa, with *S. pseudintermedius* being the most common species. The findings presented in this study indicate a variation in the burden of staphylococcal carriage among dogs by geographic distribution and an increasing trend, particularly for *S. pseudintermedius*, over the study period. Nevertheless, given the limited scope of the present study, additional research is necessary to validate these findings and ascertain the factors that influence the observed infection pattern in the study group. Additional extensive investigations that specifically examine actual infection as opposed to contamination are required to establish the burden of both infection and contaminants. Moreover, comprehensive investigations that aim to establish a relationship between the different species of *Staphylococcus* identified in this study and clinical outcomes are necessary.

## Figures and Tables

**Figure 1 animals-14-01304-f001:**
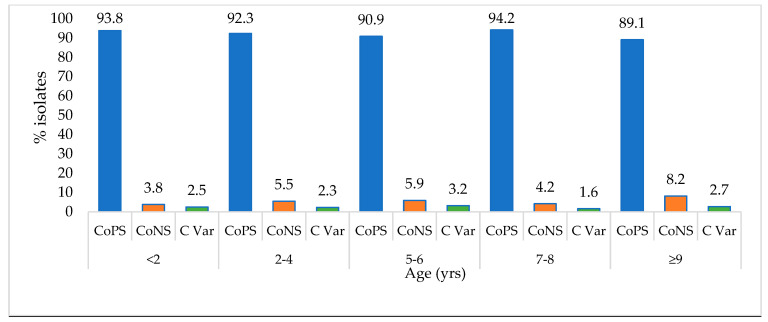
Distribution of coagulase-positive *Staphylococcus*, coagulase-negative *Staphylococcus* and coagulase-variable species in dogs among the different age groups for the period 2012–2017. CoPS, coagulase-positive *Staphylococcus*; CoNS, coagulase-negative *Staphylococcus*; C Var, coagulase-variable species.

**Figure 2 animals-14-01304-f002:**
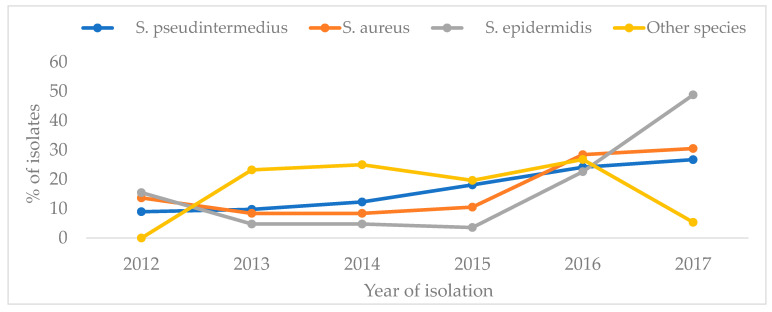
The annual distribution of *Staphylococcus* species isolated by a diagnostic laboratory in. South Africa, 2012–2017.

**Figure 3 animals-14-01304-f003:**
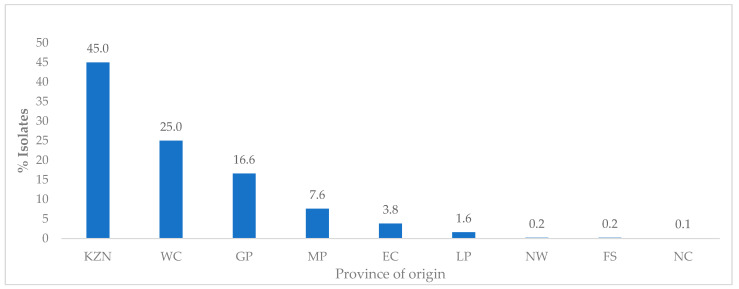
Distribution of *Staphylococcus* isolates according to the province of origin in South Africa, 2012–2017.

**Table 1 animals-14-01304-t001:** Distribution of *Staphylococcus* spp. isolated at a diagnostic laboratory between 2012 and 2017.

Organism	Frequency	Percent	95%CI
CoPS			
*S. pseudintermedius*	1398	86.0	84.14–87.58
*S. aureus*	95	5.8	4.8–7.1
CoNS			
*S. epidermidis*	84	5.2	4.1–6.4
*S. saprophyticus*	4	0.2	0.07–0.6
*S. chromogenes*	3	0.2	0.04–0.5
*S. lentus*	2	0.1	0.1–0.4
*S. felis*	1	0.1	0.1–3.4
CoPS/CoNS			
*S. species*	38	2.3	1.7–3.2
*S. schleiferi*	2	0.1	0.1–0.4

CI, confidence interval, CoPS: coagulase*-*positive staphylococci; CoNS: coagulase-negative staphylococci; CoPS/CoNS: coagulase-variable staphylococci.

**Table 2 animals-14-01304-t002:** Distribution of *Staphylococcus* spp. based on sex, age, province, season, year, and specimen type, 2012–2017.

Variable	Total No. of Isolates		CoPS		CoNS		C Var	
	*n*	%	*n*	%	*n*	%	*n*	%
Sex								
Female	762	46.8	693	90.9	50	6.6	19	2.5
Male	865	53.2	800	92.5	44	5.1	21	2.4
Year								
2012	151	9.3	138	91.4	13	8.6	0	0.0
2013	161	9.9	145	90.1	8	5.0	8	5.0
2014	197	12.1	180	91.4	8	4.1	9	4.6
2015	276	17.0	263	95.3	4	1.4	9	3.3
2016	397	24.4	366	92.2	20	5.0	11	2.8
2017	445	27.4	401	90.1	41	9.2	3	0.7
Season								
Autumn	395	24.3	357	90.4	26	6.6	12	3.0
Winter	427	26.3	390	91.3	29	6.8	8	1.9
Spring	417	25.6	379	90.9	29	7.0	9	2.2
Summer	388	24.0	367	94.6	10	2.6	11	2.8
Specimen type								
Ear	403	25.0	359	89.1	30	7.4	14	3.5
Respiratory	82	5.0	58	70.7	14	17.1	10	12.2
Skin	818	50.3	658	80.4	120	14.7	40	4.9
Urinary	175	11.0	130	74.3	35	20.0	10	5.7
Other	149	9.2	98	65.8	42	28.2	9	6.0

CoPS: coagulase-positive staphylococci, CoNS: coagulase-negative staphylococci. C Var: coagulase-variable staphylococci.

**Table 3 animals-14-01304-t003:** Correlation between age, sex, season, and specimen type.

		Specimen Type					
Variable	Level	Ear	Respiratory	Skin	Urinary	Other	*p*-Value
Sex							0.577
	Male	215	38	446	79	87	
	Female	201	32	399	67	63	
Age							0.087
	<2	24	9	86	19	13	
	2–4	117	30	212	44	40	
	5–6	75	18	173	42	35	
	7–8	81	12	166	30	26	
	≥9	106	13	181	40	35	
Season							0.222
	Autumn	98	20	199	42	36	
	Winter	106	22	214	46	39	
	Spring	104	21	210	44	38	
	Summer	97	19	194	42	36	

## Data Availability

The data that support the findings of this study are available upon reasonable request and under specific conditions. For inquiries regarding access to the data, including requests for collaboration or data sharing agreements, please contact M. Henton, Bacteriologist, at henton@vetdx.co.za. Requests are considered on a case-by-case basis, taking into consideration the nature of the request, compliance with relevant regulations, and any associated agreements or protocols.

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
