# Peer review of "A Retrospective Descriptive Study of Staphylococcus Species Isolated from Canine Specimens Submitted to a Diagnostic Laboratory in South Africa, 2012–2017"

_animals, 2024, doi:10.3390/ani14091304_

Round 1

Reviewer 1 Report

Comments and Suggestions for Authors

This manuscript performed a retrospective analysis on the Staphylococcus spp, isolates from canine specimens that were submitted to the veterinary diagnostic laboratory in South Africa in the duration of 2012-2017. The authors analyzed the Staphylococcus spp. species, distributions regarding sex and age, seasons, specimen tissues, seasons, and geographic provinces. Several critical concerns arise upon assessing this manuscript:

1. The study summarizes basic information of the Staphylococcus spp. burden in dogs in the duration of 2012-2017 in South Africa, however, it lacks the significance and novelty in the analysis to provide insights into differences among sexes, years, specimen types, and geographic regions. As a result, it does not effectively contribute to improving the detection of infections or guiding empirical treatment strategies. This issue is further underscored by the lack of substantive content in the Conclusions section.

2. The authors presented the annual distribution of Staphylococcus species in Figure 2, noting increases in S. pseudintermedius and S. epidermidis isolates. However, they fail to investigate the potential reasons behind these trends. Additionally, the figure legend lacks clarity regarding the calculation of the y-axis, impairing interpretation.

3. The authors acknowledge the need for additional investigations specifically examining contamination versus actual infection. However, if the data source is unreliable, drawing conclusions based on such analysis becomes questionable, undermining the validity of the study findings.

In conclusion, the manuscript requires substantial revisions to address the highlighted concerns and strengthen the significance of the study.

Author Response

Dear Reviewer

We tried our best to attend to all queries raised by the reviewer and they are attached below: 

Regards,

Themba Sigudu

Reviewer 2 Report

Comments and Suggestions for Authors

1. Introduction

line 54  CoPS and CoNS - abbreviations should be explained as soon as they appear in the text

2.1 Study design and setting

Whether the data came from one or many laboratories, or whether a given laboratory should not be provided (name/place etc.)?

Do none of the authors have conflicts of interest related to a given laboratory ?

How were individual isolates identified at the species level and what methodology was used to obtain species identification and test coagulase activity?

Has the same methodology for species identification and assessment of coagulase activity been applied to all isolates over the years?

Were the isolates tested for the genetic diversity of the strains present ?

2.2 Data extraction – I would like to know more data about the database – name, is the database publicly available or not etc. Apart from this, should this point not contain any notes about data management in relation to local law ?

Do the authors know/guess what may be the reason for the differences in distribution of Staphylococcus isolates according to the province of origin presented in Fig. 3 ?

I suggest adding raw data used in studies in supporting materials.

The most references are very old, if possible I suggest use of the newest literature in literature study possible.

Author Response

(The authors gave the same response as above.)

Reviewer 3 Report

Comments and Suggestions for Authors

Dear Authors, Thank you for submitting this manuscript.

I really appreciate that you have brought up the data and try to publish it as in publication. I totally agree that there is a lack of data. I will strongly encourage the authors and their co-workers to have many other constructive retrospective studies.

Please address the following point :

1) Replace ''A descriptive study" to "A Retrospective study"

2) Do you have a correlation data with the diseaes conditions ? and from that specific diseaes condition , finding of Staphylococcus?

3) Please mention how the Staphylococcus or any species were indentify, isolated and how was the data were stored?

4) What is the lab status and accreditation?

5) Please do statastical analysis, e.g. chi square test to see significance correlation in data of age, sex, season and sample type to see is there any pattern which is significant or not?

6) Did you check the data which is showing multi-etiology or multi pathogen cases?

Comments on the Quality of English Language

Minor

Author Response

(The authors gave the same response as above.)

Round 2

Reviewer 2 Report

Comments and Suggestions for Authors

line 107/108 – please modify to something like […] were extracted from the database from single veterinary diagnostic laboratory (laboratory data anonymized at his request).

lines 253, 257,279 – typo “numnber”, “shwoing”, “oon”

I believe that the lack of even basic molecular analyses is a loss for this article, as they could significantly enrich its substantive value, but I understand authors’ arguments.

In case of raw data availability – in my opinion Data Availability Statement should include all information about restriction and requirements necessary to fulfil to obtain this data.

Author Response

Reviewer: Line 107/108 – please modify to something like […] were extracted from the database from single veterinary diagnostic laboratory (laboratory data anonymized at his request).

Reply: We appreciate the suggestion from the reviewer, line 107/108 has been modified as per the reviewer suggestion and highlighted in the manuscript

Reviewer: Lines 253, 257,279 – typo “numnber”, “shwoing”, “oon”

Reply: The following typo were fixed as follows:

  • numnber (line 253) – number
  • shwoing (line 257) – showing
  • oon (line 279) – on

Reviewer: I believe that the lack of even basic molecular analyses is a loss for this article, as they could significantly enrich its substantive value, but I understand authors’ arguments.

Reply: We acknowledge the reviewer’s understanding and recognizing both the potential benefits of including molecular analyses and the rationale behind the authors' decision

Reviewer: In case of raw data availability – in my opinion Data Availability Statement should include all information about restriction and requirements necessary to fulfil to obtain this data.

Reply: The data that support the findings of this study are available upon reasonable request and under specific conditions. For inquiries regarding access to the data, including requests for collaboration or data sharing agreements, please contact Dr M. Henton, Bacteriologist, [email protected]. Requests are considered on a case-by-case basis, taking into consideration the nature of the request, compliance with relevant regulations, and any associated agreements or protocols.
